# Effectiveness of a brief intervention and text-based booster in the emergency department to reduce harmful and hazardous alcohol use: A pragmatic randomized adaptive clinical trial in Moshi, Tanzania

Catherine A. Staton[1,2*], Linda Minja[3,4], Joao Vitor Perez de Souza[1], John A. Gallis[2], Pollyana Coelho Pessoa Santos[1], Mia Buono[2], Francis Sakita[3,5], Kennedy Ngowi[4,5], Judith Boshe[3], Ashley J. Phillips[1], Joao Ricardo Nickenig Vissoci[1,2], Blandina T. Mmbaga[2,3,4,5]

1 Department of Emergency Medicine, Duke University School of Medicine, Durham, North Carolina, United States of America, 2 Duke Global Health Institute, Duke University, Durham, North Carolina, United States of America, 3 Kilimanjaro Christian Medical Centre, Moshi, Tanzania, 4 Kilimanjaro Clinical Research Institute, Moshi, Tanzania, 5 KCMC University, Moshi, Tanzania

* catherine.staton@duke.edu

## Abstract

### Background

Alcohol use contributes to over 3 million deaths annually. In Tanzania, there are no evidence-based culturally adapted interventions to address harmful alcohol use behaviors. Our hypothesis was that "*Punguza Pombe Kwa Afya Yako*" (PPKAY, Reduce Alcohol for your Health), a culturally adapted brief intervention with text-based boosters, is superior to usual care in reducing binge drinking at 3 months post discharge.

### Methods and findings

This manuscript reports. Stage 1 of our adaptive clinical trial which seeks to determine the effectiveness of the PPKAY+ booster to usual care; a subsequent stage will compare the PPKAY only to personalized and standard boosters. Adults who sought care for an acute injury at the Kilimanjaro Christian Medical Centre Emergency Department, self-disclosed alcohol use prior to the injury, scored ≥8 on the Alcohol Use Disorder Identification Test, and/or test positive by alcohol breathalyzer were offered enrollment. Participants were randomly assigned to PPKAY+ boosters (personalized or standard) or usual care at 1:1:1 allocation. Primary analyses followed the intention-to-treat principle. The PPKAY is a 15-min nurse delivered brief intervention using motivational interviewing techniques combined with a standardized or personalized text based reminder sent weekly to participants after hospital discharge and until 1 year post enrollment compared to a usual care arm. Follow-up was performed by blinded

**Data availability statement:** Data used in the preparation of this manuscript (with the exception of participants who opted out of public data sharing) are available through the National Institute on Alcohol Abuse and Alcoholism Data Archive (NIAAADA), part of the National Institute of Mental Health Data Archive (NDA). The NDA is a collaborative informatics system created by the National Institutes of Health (NIH) to support the sharing of federally funded data and accelerate research. Data for this study are stored under Collection ID: 3425 (Pro00103724) and can be accessed at https://nda.nih.gov (DOI: https://doi.org/10.15154/jeve-g615). Individual participant data, including data dictionaries, will be made available on a rolling basis beginning four months after each submission cycle. Investigators requesting access must be affiliated with an NIH-recognized research institution with an active Federalwide Assurance, and must complete a Data Use Certification approved by an authorized institutional business official with signature authority. Requests are reviewed by the Data Access Committee (DAC). This manuscript reflects the views of the authors and may not represent the opinions or views of the NIH. Analysis codes were made available at our GitHub repository (https://github.com/gemini-duke/PRACT-3Months).

**Funding:** Research reported in this publication was supported by the National Institute on Alcohol Abuse and Alcoholism of the National Institutes of Health (https://www.niaaa.nih.gov/) under award number R01AA027512 (to CAS). The funders had no role in study design, data collection and analysis, decision to publish, or preparation of the manuscript.

**Competing interests:** The authors have declared that no competing interests exist.

**Abbreviations:** AUD, alcohol use disorder; AUDIT, Alcohol Use Disorders Identification Test; BI, brief intervention; DRINC, Drinker Inventory of Consequences; KCMC, Kilimanjaro Christian Medical Centre; LMIC, low- and middle-income countries; MAR, missing at random; MNAR, missing not at random; MICE, multiple imputation by chained equations; PHQ-9, Patient Health Questionnaire-9; PMM, predictive mean matching.

outcome assessors. Our pooled intervention arms PPKAY+ boosters were compared to usual care to determine the effectiveness of the intervention in reducing the number of binge drinking days, the trial's primary outcome, in the previous 4 weeks at 3 months after discharge. A total of 1,484 patients were screened for eligibility between October 12th 2020, and on April 14th 2023. 448 patients met inclusion criteria and consented to participate. 148 were randomized to usual care, and 300 to the pooled intervention arm. Reasons for attrition included loss to follow-up ($n=69$), withdrawal ($n=6$), and deaths ($n=4$), with no differences between arms. Most participants were male (94%), from the Chagga tribe (59%) and had an average age of 36.4 years (SD 12.6) at baseline. At the 3-month follow-up, the intervention arm showed a notable reduction in mean predicted binge drinking days by 1.2 days (95% CI: [−2.3, −0.3]; $p=0.002$) compared to the usual care group in a difference-in-differences analysis. Importantly, the self-reported nature of our primary outcome introduces the potential for social desirability bias, particularly in the absence of participant blinding, and should be considered a limitation when interpreting the findings.

## Conclusion

The reduction in binge drinking behavior at 3-month follow-up compared to usual care suggests our culturally adapted intervention is an effective alcohol intervention for patients acutely injured in Tanzania. According to the adaptive study design, the next phases of the trial will continue to compare the intervention arm with a paired down version without the text messages boosters.

## Trial registration number

ClinicalTrials.gov NCT04535011

## Author summary

### Why was this study done?

- Alcohol use is a significant cause of death and disease worldwide, and Tanzania has a particularly high rate of alcohol-related problems.

- Before this study, there were a lack of proven and culturally appropriate ways to help people in Tanzania reduce harmful alcohol use.

- Although it was known that many patients injured in Tanzania use alcohol, hospitals do not routinely screen for alcohol problems or offer help.

### What did the researchers do and find?

- The researchers conducted a study with 448 adult patients seeking care for an acute injury in Tanzania who were using alcohol.

- Patients were randomly assigned to receive either the usual care at the hospital or a brief 15-min counseling session with a nurse about reducing alcohol use and follow-up reminder text messages.

- The study found that after three months, the group that received the brief counseling and text messages had a greater decrease in the number of days they engaged in binge drinking compared to usual care; on average, this was a reduction of 1.2 more binge drinking days per month.

## What do these findings mean?

- The results suggest that a short counseling session delivered by nurses, along with supportive text messages, can be a helpful way to reduce risky drinking behaviors among patients with injury in a low-resource setting like Tanzania.

- This study supports the idea of integrating such brief alcohol interventions into routine care in hospital emergency departments to reach people at a time when they might be more open to changing their drinking habits.

- Importantly, our primary outcome is self-reported and participants were not blinded, which may introduce social desirability or differential reporting bias; additionally, we enrolled all patients with AUDIT ≥8 (including those with probable dependence), and those with more severe alcohol problems may require more intensive treatment, potentially diluting the observed effect of this brief intervention.

## Introduction

Every 10 seconds a person dies from an alcohol-related cause. Alcohol contributes to 5.1% of the worldwide burden of disease and injury, and over 3 million deaths annually [1]. Africa exhibits the highest global incidence of alcohol-attributable injury deaths, with a rate of 17.1 events per 100,000 people in 2016. Additionally, the prevalence of binge drinking in Sub-Saharan African countries is among the highest globally, with 60% of current drinkers reportedly engaging in hazardous drinking behavior [1]. Tanzania is notably worse than the African average in terms of the prevalence of alcohol use disorder (AUD) (6.8% and 3.7%, respectively). Furthermore, Tanzania ranks third in the highest alcohol-attributable fractions for deaths from all causes in that region, accounting to 7.3% of all deaths in individuals over 15 years of age [1]. In Tanzania, similar to other low- and middle-income countries (LMIC) there are no evidence-based culturally adapted interventions to address harmful alcohol use behaviors.

Despite the known evidence of AUD as a public health threat in Tanzania, the lack of effective screening and treatment interventions for patients with AUD creates significant barriers to alcohol use and harm reduction [2]. Factors contributing to this cultural pattern include the normalization of alcohol-related behaviors, such as normalization and limited enforcement against drunk driving, stigmatization of alcohol consequences or alcohol care seeking behavior, early experiences with alcohol, and the locally prevalent Chagga tribe describing male alcohol use symbolizing wealth [3,4]. As a result, individuals who require help with alcohol use disorder may either not recognize the need for treatment or may fear social or cultural repercussions for seeking and obtaining treatment [3,4]. While AUD is not routinely assessed or screened for individuals attending healthcare facilities in Tanzania, our preliminary data demonstrated that 30% of patients seeking care for an injury consumed alcohol prior to injury or had an alcohol related injury [5]. Opportunistic screening and brief intervention and referral to treatment for individuals presenting to an Emergency Department is not only a World Health Organization recommended cost-effective strategy, but represents an opportunity to identify patients at high risk of alcohol related harm and intervene at a 'teachable moment' specifically when suffering an alcohol related injury [6,7].

A brief intervention (BI) is an important tool that incorporates motivational interviewing to provide individual-centered counseling for people with AUD [8,9]. The use of BIs in an ED has been shown to be a cost-effective [10] and sustainable strategy to reduce alcohol consumption [7,10,11], and number of binge drinking episodes [7]; still, there are many

opportunities to improve the implementation of a BI in the ED to promote equitable reach, adoption and intervention fidelity [12] However, most of the current literature on BIs is limited to high-income countries, with a relatively limited number of studies conducted in LMICs or underserved populations [13,14]. Furthermore, recent ED pragmatic trials have not shown a BI to be effective, likely due to implementation challenges, namely poor intervention fidelity [14,15]. For instance, the ED-SIPS pragmatic trial needed the study team to step in and conduct the interventions because clinical Emergency Department providers had low engagement; overall, 50% of their intervention group received the intervention, never mind fidelity to the tenets of behavior change were not evaluated [14]. As such, we chose a more available and trained nursing workforce to deliver the intervention and integrated a rigorous intervention fidelity program. Similarly, outside of implementation barriers, optimal components of the intervention are still in question. Some global literature has found intermittent post-intervention reminders to be helpful, while others have found limited impact [7,16]. In global settings, mobile health technology is widely available and promising, but the impact of this technology must be balanced by the process complexity, feasibility and equitable access [17,18].

Addressing the paucity of evidence-based solutions in Tanzania, a pragmatic, randomized, adaptive clinical trial (PRACT) was created to evaluate our BI's effectiveness in reducing alcohol use [19]. Herein, we will present the Stage 1 trial interim analysis results at our primary 3 months outcome follow-up in Moshi, Tanzania. We hypothesized that our nurse delivered, culturally adapted brief interventions with text based boosters would cause a greater reduction in our primary outcome (binge drinking days), and secondary outcomes (quantity or frequency of alcohol consumed, alcohol-related consequences, alcohol use disorder and depression), at 3 months post hospital discharge.

## Methods

### Ethics statement

Ethical approval was provided by the Duke University Medical Center Institutional Review Board (Pro000103724), Kilimanjaro Christian Medical University College Ethics Committee (Certificate #2457) and the Tanzanian National Institute of Medical Research (NIMR/HQ/R.8a/Vol.IX/3425). Formal written consent was obtained from all participants.

### Study design and participants

**Setting.** This study was conducted at the Kilimanjaro Christian Medical Centre (KCMC) in Moshi, in the Kilimanjaro region of Tanzania. Tanzania is one of the most populous countries in eastern Africa with over 61 million people. Despite an overall decreasing trend in alcohol use across the continent, Tanzania has the third highest and growing alcohol consumption in Africa [20]. The Kilimanjaro region has one of the highest, and rapidly increasing, reported rates of alcohol consumption per capita in the country [21]. KCMC is the regional referral center for the Kilimanjaro Region, and the emergency department typically sees around 2,000 patients seeking care for injury annually [22].

**Pragmatic study design.** The evaluation of our PPKAY intervention is through a pragmatic adaptive clinical trial. Stage 1 of this trial, reported in this manuscript, sought to determine if the PPKAY with text-based booster would reduce alcohol use at three months post-acute injury compared to a usual care arm. During Stage 2, the research question is then focused on understanding if, and which, text-based booster causes more reduction of alcohol use and alcohol related harms compared to the PPKAY alone. Patient data from Stage 1 will be used in subsequent stages to answer research questions. A priori determined stopping rules for effectiveness were used to determine when Stage 1 was completed with validation from an external Data Safety and Monitoring Board. This manuscript reports Stage 1 results comparing the PPKAY with (standard and personalized pooled) text based booster to usual care.

We adopted a pragmatic goal in our trial development and applied it to all of our decisions made during the trial development and implementation. Using the PRECIS-2 pragmatic continuum tool, we choose a typical patient population with minimal inclusion criteria, bedside recruitment during already occurring ED visits, and the research occurring in an ED where the results will be applied [23]. Similarly, we aligned the research and intervention with clinical care and expertise

so there is minimal change to implement the intervention into clinical care after the trial is complete. Given the prior literature in brief interventions and intervention fidelity, we choose to have a more rigid intervention delivery and adherence standards applied to the process but in an easily implementable, generalizable and contextually relevant way. We also chose to have few-to-no in-person follow-ups for the study, attempting to keep the care as similar to usual care as possible and we analyzed the data in an intention to treat analysis. Additional details about the protocol, study design, recruitment methods and development of the intervention are published elsewhere but will be presented in brief below and in the study protocol (S2 File) [19]. The full trial was designed and reported following the CONSORT reporting guidelines [24], and was registered on clinicaltrials.gov (NCT04535011) on August 27, 2020, and last updated on June 26, 2025.

**Participants and enrollment procedures.** Participant and enrollment procedures are described in full elsewhere [19]. In brief, participants eligible for enrollment are adult patients (≥18 years of age), who sought care for an acute injury (<24 h) at the KCMC Emergency Department. Alcohol-related criteria for inclusion comprises self-disclosed alcohol use prior to the injury, scoring ≥8 on the Alcohol Use Disorder Identification Test (AUDIT), and/or testing positive (>0.0 g/dL) by alcohol breathalyzer. Participants were required to be clinically sober at the time of enrollment and to provide informed consent. Clinicians assessed the patients' capacity to consent, considering their medical history and physical examination. Patients who arrived at the emergency department severely ill or intoxicated were repeatedly reassessed for their ability to consent. Therefore, enrollment, randomization, and intervention occurred in either the emergency department or hospital wards. The exclusion criteria included individuals who did not speak Swahili, those without access to a phone to receive text messages (this included prisoners), and those who had not been residents of East Africa for at least five years, due to the cultural adaptation aspects of our intervention. If patients met inclusion criteria, research assistants approached and enrolled participants, obtaining informed consent followed by assessment for alcohol inclusion criteria. If all inclusion and exclusion criteria are met, research assistants conducted baseline data collection.

## Randomization and masking

Patients were randomized using a computer-generated random numbers sequence, using R software. Patients were randomized into three groups (usual care, PPKAY+ personalized Booster or PPKAY+ Standard Booster) by block randomization with blocks of 12 participants at 1:1:1 allocation and will be analyzed in a 1:2 usual care to intervention fashion. Allocation envelopes were prepared by unblinded research staff before the study began. These envelopes, opaque and identical in size and thickness, contained instructions for allocation. They were sealed and stored in a locked drawer at the study site. Each envelope held a paper specifying the random group assignment for an individual participant. After screening and enrollment by blinded research assistants, unblinded clinical nurses opened these envelopes to determine group assignments and, if relevant, conducted interventions. To ensure reproducibility and quality assurance, paper enrollment packets and data collection sheets were retained. Additionally, a blinded research assistant was responsible for outcome assessments during follow-up.

## Procedures

**Brief intervention.** Participants assigned to one of the PPKAY (*Punguza Pombe Kwa Afya Yako*, "Reduce Alcohol for Your Health") arms engaged in a 15-min discussion led by a nurse [24]. This intervention, adapted linguistically and culturally from "Screening and Brief Intervention for Unhealthy Alcohol Use in the Emergency Department," [25] involves a four-step dialogue: (1) addressing the topic of alcohol, (2) offering feedback, (3) bolstering motivation, and (4) providing guidance and negotiating. While a comprehensive description of the intervention's cultural adaptation and the development and testing of text based booster messages is detailed elsewhere [19,26], in summary, PPKAY comprises a brief nurse-conducted conversation about alcohol use and its personal impact on the participant. This is followed by a discussion on readiness to change, guided by motivational interviewing techniques, and goal-setting. Adaptations for the local context included the use of the most appropriate relevant vocabulary in Tanzanian Swahili, highlighting the most

common complications of alcohol use in the Kilimanjaro region, clear emphasis on the link between alcohol use and its consequences, and an obligation to advise against drink driving for all participants, among others [26].

**Text-based booster messages.** The development and feasibility evaluation of the text based booster messages is reported elsewhere but in brief below [18]. Participants in the PPKAY+ Standard Booster group underwent the PPKAY intervention as previously described and also received weekly text messages for one year [18,19]. An illustrative standard booster text message, in English, is: "*Set a goal and try to reduce your drinking this weekend. You can do it!*". In contrast, those in the PPKAY+ Personalized Booster group not only participated in the PPKAY process but also received customized text messages. Clinical nurses documented each patient's specific goals and reasons for wanting to change during the PPKAY discussion. These details were then incorporated into the motivational texts [19]. The data manager reviewed these texts before they were programmed into the short message service system. An example of such a personalized text message is: "*Stay focused. Make a plan. Write down the reasons you want to drink less. Remember your motivation to spend more time with your family*" if a patient had expressed a desire to reduce drinking to spend more time with family.

**Usual care condition.** Stage 1 PRACT goal is to understand the benefits of PPKAY with text-based booster, compared with the current standard practice or usual care (UC). Currently, standard practice at KCMC for patients with injury does not include screening or testing for alcohol use, alcohol use disorder, or its consequences, nor education on the consequences of alcohol use.

**Intervention fidelity.** The PPKAY intervention was implemented pragmatically with clinical nurses conducting the intervention. Repeat training sessions were implemented to ensure enough clinical nurses were trained on how to conduct the intervention during the study period. Similarly, all interventions were recorded and 10% of them were evaluated using an adapted, piloted and validated intervention checklist by a Swahili speaking researcher. Nurse interventionists were assessed for each intervention performed until their 5th, in order to receive feedback on their performance and possible improvement. Then, 10% of all interventions were evaluated for quality by both a researcher and our Tanzanian/Swahili speaking Psychiatrist to ensure fidelity. Recurrent gaps in fidelity or areas of improvement found were then integrated into monthly feedback to nurse interventionists as a whole, one-on-one as needed and into our training curriculum for future nurse training processes.

**Outcomes measures.** All outcomes were assessed at 3 months post-discharge, as commonly shown in the literature [27]. The primary outcome was the mean predicted change in the number of binge drinking days (count) in the previous 4 weeks, as proposed in the trial protocol registered at NCT04535011. The definition of our primary outcome in the published trial protocol is slightly different. The change from proportion of binge drinking to number in binge drinking days was due to the rarity of the event [19]. This change was approved by our Advisory and Data Safety Monitoring Boards and implemented in our Statistical Analysis Plan before any analysis had been performed. In this study, binge drinking is defined as having at least 4 or 5 standard drinks in a single day for women and men, respectively. In our prior data, binge drinking carries an increased risk of injury requiring healthcare services [5]. Secondary outcomes measures included alcohol-related consequences (continuous, measured by The Drinker Inventory of Consequences [DrinC]) [28], number of standard drinks (count) consumed, number of drinking days (count), AUDIT score (continuous), [29] and depressive symptoms (continuous, measured by the PHQ-9) [30]. The DrinC and AUDIT questions refer to a three-month time window; number of drinking days and amount of alcohol to a four-week period; and, PHQ-9 to a two-week window. All outcomes were self-reported and collected via phone calls during the standard follow-up windows using a timeline follow back procedure [31]. Alcohol consumption patterns were captured by asking participants what they drank and how much (e.g., 2 glasses of wine), and then converted into international standard drinks by trained research assistants. All participants received 5,000 Tanzanian shillings or the equivalent of about $2.00 US dollars for participating in the research follow-ups, delivered by the MPesa system, or a nationally available and ubiquitous phone managed finance system.

**Sample size estimation.** The sample size was determined based on detecting a 35% reduction in the number of binge drinking days per month in the primary outcome across treatment arms. Based on the literature, we assumed a standard deviation of 3.4, and a 35% difference translates into a difference of about 1 binge drinking day, assuming a baseline average of four binge drinking days. To detect this difference with 80% power and a two-sided alpha of 0.05, we estimated that 164 participants per arm would be needed. Given the intent of our adaptive design to reduce the amount of patients exposed to Usual Care, we planned three interim analyses (at 20, 65, and 100% enrollment and complete follow-ups), using the O'Brien-Fleming (OBF) alpha-spending function to preserve the overall type I error rate. Nominal alpha thresholds for each stage were determined using the ldBounds and gsDesign packages in R. The current analysis is the report of the second planned interim analysis, after approximately two-thirds of the planned sample size was reached. At this stage, the nominal significance level required for early stopping was adjusted per the OBF boundary, and the results met the creator set a priori.

## Statistical analysis

Patient demographics were compared across all study arms. Descriptive data was reported as means, standard deviations, medians, quartiles, and frequencies. Outcome data are reported as predicted means with confidence intervals for each time point. Using an intention to treat analysis, we tested the primary hypothesis that participants enrolled in the intervention arms would have a larger reduction in binge-drinking events in the past month, compared to those enrolled in Usual Care. We used a constrained longitudinal analysis approach to address potential outcome differences at baseline between arms [32]. This approach allows for the prediction of follow-up scores after controlling for participants' baseline characteristics. The model was fitted using a mixed-effect zero-inflated negative binomial method with a log link function, given the count nature and over dispersion of alcohol consumption data for primary outcome, as well as excess zero count due to participants who do not binge drink. Differences in the number of drinking days and number of drinks consumed by enrollment arms were analyzed using mixed-effect negative binomial models. Linear mixed effects models were used to assess changes in the AUDIT, DrInC, and PHQ-9 scores. Using the constrained longitudinal approach, we computed the differences between arms of the changes from baseline to follow-up. All analyses were conducted according to an intention-to-treat approach. The average predicted change in the outcome measures from baseline to 3 month follow-up was calculated and a bootstrapping approach with 1,000 replications was used to calculate the 95% confidence intervals. The comparison of the treatment arms is made with a difference in difference, with a bootstrapping approach with 1,000 replications to calculate the confidence intervals. Results for the models are reported as predicted means and differences in means. Analysis was conducted using the R Language and environment for statistical computing (version 4.3) [33]. More details on the analytical procedures are available in the Supplementary Material (S3 File).

**Missing data.** Presence and patterns of missing data were explored using descriptive statistics and UpSet plots. We analyzed the association of our patient characteristics with the presence of missing data and, although missing not at random (MNAR) cannot be empirically ruled out, we assume a missing at random (MAR) mechanism conditional on the observed covariates included in the imputation model, as is standard practice in multiple imputation (Table C in S1 File). To address missingness, we implemented multiple imputation by chained equations (MICE) using the mice package in R [34]. Imputation of continuous variables was conducted using Predictive Mean Matching (PMM), a semi-parametric method that preserves the distribution of the observed data by selecting observed values from similar cases rather than drawing synthetic estimates. Imputation models included all patient demographic variables and outcome measures. More details on the imputation procedure, including model specifications and variables included, are provided in the Statistical Appendix. A total of 100 imputed datasets were generated. For each dataset, up to 50 iterations were performed to ensure convergence of the imputation model, preceded by 100 burn-in iterations. Analyses were conducted separately across the imputed datasets and subsequently pooled using Rubin's rules to obtain final estimates. The primary modeling results

presented in this manuscript are based on these pooled analyses using the imputed datasets. In addition, we also report the results for the complete case analysis in Table D in S1 File.

## Sensitivity analysis

We conducted sensitivity analyses for both primary and secondary outcomes to assess the robustness of findings under alternative model specifications, different handling of missing data, and exclusion of statistical outliers. These included complete case analyses, non-parametric approaches, and alternative count-data models that account to varying degrees for overdispersion and excess zeros. Full specifications and results for these analyses are provided in the S1 File (Tables D–N and Fig A).

## Results

Participant enrollment started on October 12th 2020, and on April 14th 2023, we conducted our second interim look at our data; a total of 1,484 participants were screened for eligibility, with 448 meeting the inclusion criteria and consenting to participate in the study (Fig 1). Of the 1,034 not meeting eligibility criteria, 1,032 did not meet one of the alcohol eligibility criteria (self-report of alcohol use, AUDIT ≥ 8 or breathalyzer > 0.00), and 2 declined to participate after beginning screening procedures. The criteria for enrollment included a breathalyzer test, which was the least frequently positive (9, 2.5%), with no patients testing positive solely on this criterion (Table A in S1 File). Of the other alcohol enrollment criteria, AUDIT scores ≥8 (258, 70.1%) was similar to self-reported alcohol use before injury (263, 71.3%). From the 448 enrolled patients, 148 were randomized to usual care and 300 were randomized to the intervention arms. Seven patients did not receive the allocated intervention but were still factored into the analysis under the intention-to-treat principle. At the 3-month follow-up, 123 participants completed follow-up in the usual care arm, and 246 participants completed follow-up in the intervention arm. Reasons for attrition included loss to follow-up ($n=69$), withdrawal ($n=6$), and deaths ($n=4$), with no differences between arms (Fig 1). Descriptive data for patients lost to follow-up is available in the Table B in S1 File. Other than the 4 patient deaths, which are believed to be unrelated to study activities, no adverse events occurred during

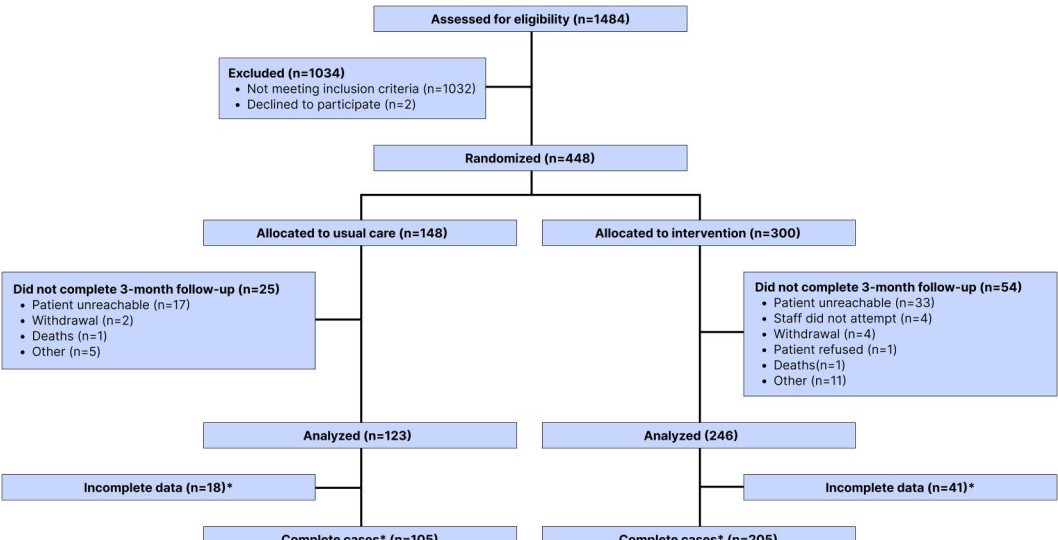

**Fig 1. Flow of participants through this randomized controlled trial.** Alcohol-related criteria for inclusion comprises self-disclosed alcohol use prior to the injury, scoring ≥8 on the AUDIT, and/or testing positive (>0.0 g/dL) by alcohol breathalyzer. *The primary and secondary outcome results presented in this study represent data imputed using MICE. Complete case analysis is reported in the Table D in S1 File.

this study period. The trial was initiated during the COVID-19 epidemic, and our team adhered to Tanzanian and hospital based national guidelines including personal protective equipment and not allowing persons over 65 to present to work. During a COVID-19 wave, with reduced staff, our research team missed 9 of the follow-ups at the 3 month window, of which 8 happened to be in the intervention arms. The data used for Tables 1 and 2 reflect 369 patients that completed 3-months follow-up. Outcome data was available for 71% (105/148) and 68% (205/300) of the participants allocated to the usual care and intervention groups, respectively. Missing data was addressed by using multiple imputation by chained equations. In addition, we also report the results for the complete case analysis in the Table D in S1 File.

The two groups had generally similar characteristics at baseline in terms of age, self-reported sex, education, and both individual and household income (Table 1).

**Table 1. Baseline demographics information of the study sample.**

| Characteristic | Overall N = 369[a] | Usual care, N = 123[a] | Intervention, N = 246[a] |
|---|---|---|---|
| **Age** | | | |
| Mean (SD) | 36.4 (12.6) | 35.3 (12) | 37 (12.9) |
| Median (Q1, Q3) | 34 (27, 43) | 34 (26.5, 41.5) | 34 (27, 44) |
| **Self-reported sex** | | | |
| Male | 346 (94%) | 114 (93%) | 232 (94%) |
| Female | 23 (6.2%) | 9 (7.3%) | 14 (5.7%) |
| **Tribe** | | | |
| Chagga | 216 (59%) | 73 (59%) | 143 (58%) |
| Other | 116 (31%) | 37 (30%) | 79 (32%) |
| Pare | 37 (10%) | 13 (11%) | 24 (9.8%) |
| **Years of education** | | | |
| Mean (SD) | 9 (3.4) | 9.5 (3.5) | 8.7 (3.3) |
| Median (Q1, Q3) | 7 (7, 11) | 7 (7, 11) | 7 (7, 11) |
| Missing | 1 | 0 | 1 |
| **Employment** | | | |
| Skilled-employment | 101 (29%) | 23 (20%) | 78 (33%) |
| Unskilled-employment | 96 (27%) | 32 (28%) | 64 (27%) |
| Farmer (self-employed) | 84 (24%) | 30 (26%) | 54 (23%) |
| Professional | 42 (12%) | 21 (18%) | 21 (8.8%) |
| Other | 27 (7.6%) | 10 (8.6%) | 17 (7.1%) |
| Farmer (employee) | 4 (1.1%) | 0 (0%) | 4 (1.7%) |
| Missing | 15 | 7 | 8 |
| **Monthly total household income (Tz Shilling)** | | | |
| Mean (SD) | 525,614.5 (822,356.5) | 570,697.7 (901,983.2) | 505,100.5 (785,134.4) |
| Median (Q1, Q3) | 300,000 (150,000, 505,000) | 300,000 (177,500, 600,000) | 300,000 (150,000, 500,000) |
| Missing | 94 | 37 | 57 |
| **Monthly personal income (Tz Shilling)** | | | |
| Mean (SD) | 377,451.2 (566,478.6) | 392,747.7 (584,063.3) | 369,626.7 (558,478.9) |
| Median (Q1, Q3) | 240,000 (100,000, 400,000) | 250,000 (150,000, 450,000) | 240,000 (100,000, 400,000) |
| Missing | 41 | 12 | 29 |

[a]n/N (%).

Note: The intervention group is a pooled sample combining participants receiving standard and personalized text messages.

**Table 2. Primary and secondary outcomes for baseline and 3 months for usual care and intervention.**

| Characteristic | Baseline | | | 3 Months | | |
|---|---|---|---|---|---|---|
| | N = 369[a] | Usual care N = 123[a] | Intervention N = 246[a] | N = 369[a] | Usual care N = 123[a] | Intervention N = 246[a] |
| Binge drinking days | | | | | | |
| Mean (SD) | 2.8 (5.8) | 2.2 (5.4) | 3.0 (6.0) | 0.5 (3.0) | 0.8 (3.3) | 0.4 (2.9) |
| Median (Q1, Q3) | 0.0 (0.0, 3.0) | 0.0 (0.0, 1.5) | 0.0 (0.0, 3.0) | 0.0 (0.0, 0.0) | 0.0 (0.0, 0.0) | 0.0 (0.0, 0.0) |
| Min, Max | 0.0, 28.0 | 0.0, 28.0 | 0.0, 28.0 | 0.0, 28.0 | 0.0, 28.0 | 0.0, 28.0 |
| Missing | 28 | 7 | 21 | 34 | 12 | 22 |
| N binge drinking days | | | | | | |
| None | 217/ 369 (59%) | 77/ 123 (63%) | 140/ 246 (57%) | 344/ 369 (93%) | 110/ 123 (89%) | 234/ 246 (95%) |
| One | 41/ 369 (11%) | 17/ 123 (14%) | 24/ 246 (9.8%) | 9/ 369 (2.4%) | 4/ 123 (3.3%) | 5/ 246 (2.0%) |
| More than 1 | 111/ 369 (30%) | 29/ 123 (24%) | 82/ 246 (33%) | 16/ 369 (4.3%) | 9/ 123 (7.3%) | 7/ 246 (2.8%) |
| Drinking days | | | | | | |
| Mean (SD) | 7.6 (7.6) | 7.4 (8.1) | 7.7 (7.3) | 1.9 (5.2) | 2.9 (6.7) | 1.4 (4.2) |
| Median (Q1, Q3) | 5.0 (2.0, 10.0) | 4.0 (2.0, 9.5) | 5.0 (3.0, 10.0) | 0.0 (0.0, 1.0) | 0.0 (0.0, 1.0) | 0.0 (0.0, 0.0) |
| Min, Max | 0.0, 28.0 | 0.0, 28.0 | 0.0, 28.0 | 0.0, 28.0 | 0.0, 28.0 | 0.0, 28.0 |
| Missing | 28 | 7 | 21 | 34 | 12 | 22 |
| Number of drinks | | | | | | |
| Mean (SD) | 36.9 (54.3) | 33.6 (56.5) | 38.6 (53.1) | 7.9 (28.1) | 10.4 (27.1) | 6.6 (28.6) |
| Median (Q1, Q3) | 20.0 (6.3, 42.0) | 17.5 (3.8, 36.5) | 22.0 (7.0, 46.0) | 0.0 (0.0, 1.0) | 0.0 (0.0, 6.0) | 0.0 (0.0, 0.0) |
| Min, Max | 0.0, 456.0 | 0.0, 456.0 | 0.0, 315.0 | 0.0, 261.0 | 0.0, 157.0 | 0.0, 261.0 |
| Missing | 28 | 7 | 21 | 34 | 12 | 22 |
| PHQ-9 score | | | | | | |
| Mean (SD) | 3.0 (3.8) | 2.7 (3.5) | 3.2 (3.9) | 3.9 (4.3) | 3.6 (4.4) | 4.1 (4.3) |
| Median (Q1, Q3) | 2.0 (0.0, 4.0) | 2.0 (0.0, 4.0) | 2.0 (1.0, 4.0) | 3.0 (0.0, 6.0) | 2.0 (0.0, 6.0) | 3.0 (1.0, 6.0) |
| Min, Max | 0.0, 24.0 | 0.0, 17.0 | 0.0, 24.0 | 0.0, 22.0 | 0.0, 22.0 | 0.0, 18.0 |
| Missing | 11 | 6 | 5 | 7 | 4 | 3 |
| AUDIT score | | | | | | |
| Mean (SD) | 12.9 (7.4) | 12.3 (6.5) | 13.2 (7.8) | 4.0 (4.6) | 4.1 (4.3) | 4.0 (4.7) |
| Median (Q1, Q3) | 11.5 (8.0, 18.0) | 11.0 (8.0, 17.0) | 12.0 (8.0, 18.0) | 4.0 (0.0, 6.0) | 4.0 (0.0, 6.0) | 4.0 (0.0, 5.0) |
| Min, Max | 1.0, 38.0 | 1.0, 33.0 | 1.0, 38.0 | 0.0, 32.0 | 0.0, 23.0 | 0.0, 32.0 |
| Missing | 1 | 0 | 1 | 6 | 1 | 5 |
| DrInC | | | | | | |
| Mean (SD) | 14.2 (16.4) | 13.8 (15.5) | 14.4 (16.8) | 2.5 (7.9) | 2.8 (7.9) | 2.3 (7.9) |
| Median (Q1, Q3) | 9.0 (3.0, 20.0) | 9.5 (3.0, 19.0) | 8.0 (2.0, 20.0) | 0.0 (0.0, 1.0) | 0.0 (0.0, 1.0) | 0.0 (0.0, 0.0) |
| Min, Max | 0.0, 96.0 | 0.0, 96.0 | 0.0, 93.0 | 0.0, 62.0 | 0.0, 56.0 | 0.0, 62.0 |
| Missing | 40 | 17 | 23 | 89 | 37 | 52 |

[a]n/N (%),

Note: The intervention group is a pooled sample combining participants receiving standard and personalized text messages.

PHQ-9 = Patient Health Questionnaire-9.

AUDIT = Alcohol Use Disorder Identification Test.

DrInC = The Drinker Inventory of Consequences.

At 3-month follow-up, both the usual care and intervention groups had a decrease in the number of binge drinking days (See Table 2). The mean number of binge drinking days in the intervention group decreased by approximately 2.6 days (from 3 to 0.4), while the usual care group experienced a decrease of 1.4 days (from 2.2 to 0.8). This observed reduction across both study arms highlights the overall trend towards reduced alcohol use during the follow-up period.

We observed a significant reduction in binge drinking days for the intervention group compared to the usual care group (Fig 2). The change in the mean number of binge drinking days was estimated to be −2.9 (95% CI [−3.9, −2.2]) days in the intervention group, and −1.7 (95% CI [−2.2, −1.3]) days in the usual care group. The difference in differences between the groups of −1.2 (95% CI [−2.3, −0.3], p = 0.002) represented, on average, 71% higher reduction in binge drinking days for the intervention groups. Fig 2 shows the changes in binge drinking days for both groups from baseline to the 3-month follow-up. The results met our boundary for completing this stage of the trial. To assess robustness, we conducted several sensitivity analyses: (i) complete-case analyses versus the multiple imputed primary analysis, (ii) exclusion of statistical outliers in both complete-case and imputed datasets, and (iii) complementary analytical approaches including a nonparametric Wilcoxon rank-sum test of change scores and inspection of coefficients from the prespecified zero-inflated negative binomial model. In the complete cases, the difference in differences (DID) was −1.4 days (95% CI [−1.7, −1.0]; p = 0.004; Table D in S1 File). After excluding extreme outliers, results were similar: complete cases DID −1.0 days (95% CI [−1.3, −0.7]; Table E in S1 File) and multiply imputed data DID −0.92 days (95% CI [−1.45, −0.47]; Table F in S1 File). Nonparametric Wilcoxon rank-sum tests comparing change scores also favored the intervention across all samples (p = 0.005–0.015; Table G in S1 File). In the prespecified model, model coefficients for the longitudinal variation in the outcome was significant (IRR 0.25, 95% [CI 0.09, 0.69]; p = 0.008; Table H in S1 File), consistent with greater reductions in the intervention arm. Individual-level change distributions are shown in Fig A in S1 File. Full model specifications and estimation procedures are detailed in the Supplementary Material (S2 and S3 Files).

For our secondary outcomes, drinking days and the number of drinks decreased in both arms during the study period, with substantial decreases in the intervention group. The mean number of drinking days was reduced from 7.7 to 1.4 days in the intervention group, compared to a decrease from 7.4 to 2.9 days for usual care (Table 2). Similarly, the mean number of drinks consumed in the last four weeks reduced from 38.6 to 6.6 and from 33.6 to 10.4, in the intervention and control groups, respectively (Table 2). The predicted difference-in-differences in drinking days between the intervention and usual care groups was −1 day (95% CI [−2.3, 0.4]), and for number of drinks was −11.1 (95% CI [−2.3, −0.3]) in models fit using a multiple imputation approach (Table 3). The confidence intervals for both secondary outcomes suggest a wider

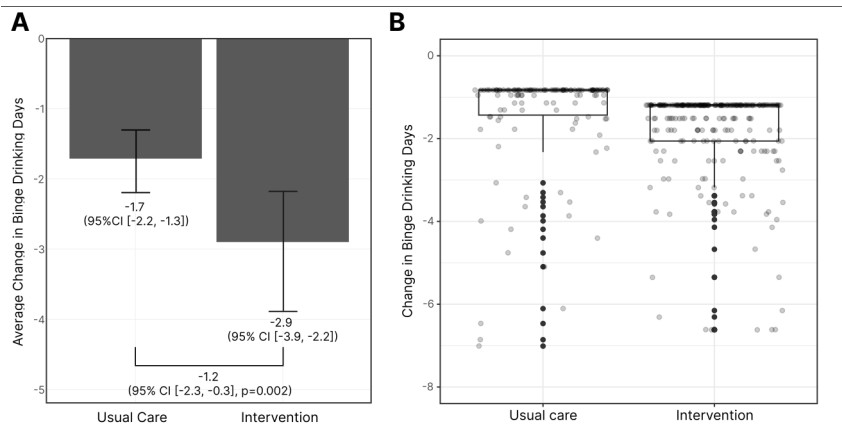

**Fig 2. Reduction in the number of binge drinking days from baseline to 3-months follow up for intervention and usual care and difference in difference between groups.** Panel **A** displays average predicted change in binge drinking days. Vertical error bars represent 95% confidence intervals. Panel **B** displays a Boxplot of the within-group differences highlighting the individual-patient values (Y axis truncated to −8 for ease of visualization). Box boundaries represent the first and third quartiles, and vertical whisker represents the minimum values observed, excluding outliers. In the case of this variable, the maximum value was equivalent to the median, thus the maximum and median are not differentiated from the first quartile value presented in the figure. Refer to Fig A in S1 File for the visualization of the observed changes in binge drinking as well as complete case vs. multiply imputed differences with and without y axis truncation.

**Table 3. Predicted means and mean predicted difference with respective confidence intervals and *p* values for secondary study outcomes.**

| Secondary outcome | Arm | Predicted means | | Difference in means (95% CI)[a] | Difference in differences (95% CI)[a] | *p* value |
|---|---|---|---|---|---|---|
| | | Baseline | 3 months | | | |
| Drinking days | Intervention | 8.1 | 0.8 | −7.2 (−8.1, −6.4) | −1 (−2.3, 0.4) | <0.001[b] |
| | Usual care | 8 | 1.8 | −6.2 (−7.4, −5.1) | | |
| Drinking amount | Intervention | 51.5 | 2.8 | −46 (−56.5, −37.4) | −11.1 (−23, −0.3) | <0.001[b] |
| | Usual care | 41.3 | 6 | −34.9 (−41.7, −28.8) | | |
| AUDIT | Intervention | 13 | 3.9 | −9.1 (−9.5, −8.7) | −0.3 (−0.9, 0.2) | 0.54[c] |
| | Usual care | 12.8 | 4 | −8.8 (−9.2, −8.3) | | |
| DrInC | Intervention | 14.7 | 3.5 | −10.9 (−11.7, −9.9) | −0.4 (−1.8, 1) | 0.61[c] |
| | Usual care | 14.8 | 4.6 | −10.5 (−11.6, −9.2) | | |
| PHQ9 | Intervention | 3.1 | 4.1 | 1 (0.8, 1.2) | 0.4 (0.1, 0.7) | 0.39[c] |
| | Usual care | 3 | 3.6 | 0.6 (0.4, 0.9) | | |

[a]Positive and negative values reflect the reduction or increase from predicted values at baseline compared to follow-up, respectively.

[b]Mixed-effect negative binomial models.

[c]Linear mixed-effect models.

*p* values represent the significance of the difference in differences.

PHQ-9 = Patient Health Questionnaire-9.

AUDIT = Alcohol Use Disorder Identification Test.

DrInC = The Drinker Inventory of Consequences.

range of potential intervention effects, with a stronger to modest association with drinking frequency and quantity, though the range of plausible effects remains wide. Sensitivity analyses using additional analytical approaches (Tables I–K for drinking days; Tables L–N for number of drinks in S1 File) produced results with consistent directionality and magnitudes.

At the 3-month follow-up, the intervention group had lower predicted AUDIT and DrInC scores than the usual care group, with small between-group differences of −0.3 (95% CI [−0.9, 0.2]) and −0.4 (95% CI [−1.8, 1]) in the models fit using a multiple imputation approach, respectively. For PHQ-9 scores, both groups experienced a small increase from baseline, with a between-group difference of 0.4 (95% CI [0.1, 0.7]). We observed a small difference in depression scores supporting a slight increase in depression post intervention. However, this result was not consistent during sensitivity analysis (see Appendix). Across all secondary outcomes, the confidence intervals included values close to or spanning the null, suggesting that any true effects may be small or absent. Given that the study was not powered to detect differences in these outcomes, the results should be further verified. These exploratory findings could guide the design and prioritization of future studies rather than serve as definitive evidence of intervention impact.

## Discussion

This study evaluates the effectiveness of a culturally- adapted BI, 'Punguza Pombe Kwa Afya Yako'/'Reduce Alcohol for your Health' (PPKAY) with text-based booster in reducing alcohol use in Moshi, Tanzania. To our knowledge, it is the first study to present a pragmatic, randomized, adaptive clinical trial evaluating an alcohol harm reduction intervention in a LMIC. Our outcome was the change in the number of binge drinking days between intervention and usual care at a 3-month follow-up. The results revealed a significantly larger reduction in binge drinking days, drinking days, and number of drinks per drinking event in the intervention arm. A zero-inflation component and longitudinal constraint approach were added to our model to account for the rarity of our outcomes and the slightly higher baseline drinking in the intervention group. We found the intervention group had an average reduction of 1.2 binge drinking days in the last four weeks compared to the control. Our sensitivity analysis showed minimal variation in the estimated results. However, the wide

standard deviations for the differences between groups and the rarity characteristic of the outcome reflect a wider uncertainty on the range of the effect, varying from approximately 0.3 to 2.0 days per month. Additionally, the intervention group had lower drinking days and number of drinks at 3 months compared to usual care. While these estimates suggest an additional potential benefit of the intervention, these outcomes were secondary, and the study was not powered to detect differences in them.

Previous literature has explored the efficacy of BIs and pragmatic randomized clinical trials in either high-income countries or in primary care settings [14,35–37]. In contrast to many of these trials, our present study investigates brief interventions initiated in an emergency department of a low-income country. Our primary focus of patients with injuries was due to the high burden of alcohol use among this sample at KCMC, the strength of the literature for a BI in the care seeking injury population, and being cognizant of the burden this screening and BI would create in this nascent Emergency Department; there is interest in expanding this treatment option to the whole Emergency Department population. Especially in LMIC, our Emergency Department setting is optimal to centralize scarce expertise and human resources while building these tools, and referral systems to then support more general health system implementation. Prior Emergency Department pragmatic trials had significant intervention fidelity challenges. For instance, the ED-SIPS trial in the United Kingdom found no difference in the BI arm when compared to 5-min advice or simple clinical feedback and leaflet material [14]. Unfortunately, the ED-SIPS trial had limited intervention fidelity, with as few as 50% of the patients in the brief counseling arm receiving the intervention, the research team conducting the intervention due to a lack of provider engagement, a lack of usual care comparator, and attrition during follow-up, all of which may explain the null results [14,38]. To account for implementation challenges, our intervention and implementation strategy emphasized cultural adaptations and focused on intervention fidelity. Our rigorous fidelity program includes an in-depth onboarding process with feedback for each interventionist until they complete 5 quality interventions, a continuous quality control process, including a checklist-based audit and feedback of 10% of the interventions by our Swahili speaking Psychiatrist investigator, to maintain high fidelity and address any deviations promptly.

Our team made specific cultural and resource-appropriate adaptations that we wish to note. First, we included all patients with an AUDIT ≥ 8. In most high-resource settings, a brief intervention is typically only indicated for those with AUDIT 8–18, as those with alcohol dependence (AUDIT > 18) might require more intensive services. That said, the ED-SIPS trial and others also included all drinkers AUDIT ≥8, citing literature that those with alcohol dependence might benefit from a brief intervention [14,39]. In our setting, there is essentially no access to other alcohol treatment services, and thus, withholding this intervention would be unethical. As such, our inclusion of all those with AUDIT ≥8 allows us to understand a pragmatic effect of our intervention, and a planned moderator and mediator analysis will help to narrow our population if appropriate or needed.

Second, we incorporated three alcohol-related enrollment characteristics in our study: an AUDIT score of 8 or higher, self-report of alcohol use prior to injury, and breathalyzer testing. Our results indicated that no patient tested positive by breathalyzer alone, without also meeting the AUDIT or self-reported alcohol use criteria. Similarly, prehospital times to reach care in Tanzania can be extensive; 54% of patients with injury at KCMC presented more than 4 hours after the incident [40], underscoring the challenges posed by no prehospital care system, prolonged distances to hospitals, and a complex referral system for trauma care typical of low-resource settings. Given these points, while breathalyzer is used in high-resource settings, we posit that it provides limited to no added benefit for implementation of alcohol harm reduction interventions in low-resource settings [36]. As such, we also included a self-reported measure of alcohol prior to injury as a potential alternative to the breathalyzer, as the literature cites an injury attributed to alcohol increases their risk and the chances of behavior change. Further subgroup analyses are planned to understand any possible associations with outcomes.

Our results show that a culturally adapted brief intervention with fidelity monitoring was associated with greater short-term reductions in binge drinking than usual care among patients with injury. Our Stage 1 PRACT results have

demonstrated effectiveness compared to usual care at 3 months and support the implementation of this intervention. Our subsequent stage results, along with our longer-term outcomes will be further analyzed in a subsequent manuscript in order to inform implementation planning and optimal follow up timing in this setting. Further evaluation of our mediators and moderators can help delineate the optimal population in order to inform dissemination strategies for our low resource setting.

There are important limitations which should be considered when evaluating our results. First, as mentioned, we have included all eligible patients with AUD scores ≥8; if those with alcohol dependence do need more intense treatment, it is possible that alcohol-dependent patients' minimal behavior change is masking a potentially more impactful effect on those with harmful and hazardous drinking. Further evaluation with mediator and moderator analyses are planned and include assessment of the highest-risk groups of our population. Because of this pragmatic approach and our limited exclusion criteria, we expect our result to be underestimated overall.

We are using primarily self-reported alcohol use tools in our enrollment and outcome assessments. Self-report measures are limited by definition but also provide a more patient centered evaluation of the intervention. Our team has undergone extensive prior work to define the percent alcohol per alcohol unit, where possible, understand the quantification of a 'standard drink' and train our blinded team members to calculate this based on patient self-reporting of alcohol use using a standard timeline-follow-back method [26]. These tools are generalizable, sustainable and appropriate in our low-resource setting. Similarly, while we have identified a strong stigma against alcohol-related consequences, our decade-long work with this population has taught us how to approach patients and handle stigmatized topics.

While 82% of the acute injury population at KCMC have cell phones, the exclusion of individuals without mobile phones may have disproportionately impacted those with more severe injury (who lost their phone during the injury) and or socioeconomically disadvantaged groups. The future stage of PRACT will compare PPKAY alone to PPKAY with SMS to guide decisions on this component, and our enrollment required phone contact for the SMS intervention and follow up purposes.

We see a variation between groups at baseline, for binge drinking days, drinking days and number of drinks in the intervention group than the usual care. Variations on self-reported outcomes are common in behavior-change trials. We believe these are attributable to the randomness of the trial design since our participant demographic characteristics showed balance indicating the appropriateness of randomization. However, this is a common and known limitation and we anticipated this variation could exist and planned our analysis to address this limitation with our analytical approach. Since we are pragmatically focused on improving access to alcohol treatment, minimizing the number of people in the usual care group while still obtaining effectiveness data was a priority. The reliance on self-reported measures introduces the potential for social desirability bias, particularly in the absence of participant blinding, which could lead to differential reporting between arms. Nonetheless, because we did not include a formal negative control outcome, future studies should consider incorporating such measures or objective biomarkers to further validate these findings. We believe that examining Stage 2 of PRACT (PPKAY alone versus PPKAY with booster) and Stage 1 for other follow up time periods data may help us understand more of this phenomenon.

We observed a reduction of alcohol use regardless of treatment allocation. This is observed in other alcohol intervention trials, especially those conducted in emergency care settings, due to contextual factors such as trauma exposure or the acute care experience [15,27,37,41–43]. Similarly, because of the rarity of our outcome, our observed mean values are skewed towards zero which can also inflate the difference in the observed means from baseline to follow-up, potentially distorting the interpretation of the intervention's effectiveness. We adopted an analytical approach that is designed to handle this type of data and report our effect as predicted means to provide the adjusted estimate.

In this pragmatic, randomized, adaptive clinical trial demonstrated the effectiveness of our culturally adapted brief intervention, Punguza Pombe Kwa Afya Yako (PPKAY). Compared to usual care, PPKAY with mobile health-based boosters reduced binge drinking by an average of 1.2 days more per four weeks. According to the adaptive study design, the next phase of the trial will carry on the intervention arms and compare it with a paired down version of the intervention, without

the text message boosters. Binge drinking and other high-risk alcohol use are common causes of injury and violence which cause extensive individual, health system and societal burdens. This 15-min nurse-delivered culturally- adapted intervention can effectively reduce binge drinking, and the quantity and frequency of alcohol use at 3 months post-acute injury. Further delineation of optimal implementation and population are warranted.

## Supporting information

**S1 File. Supplementary tables and figures.** Fig A: Observed and predicted changes in binge drinking days using multiply imputed datasets, with and without statistical outliers. Table A: Enrollment criteria for patients that completed 3 months follow-up. Table B: Demographic and outcome characteristics for patients lost to follow-up at 3 months by study arm. Table C: Descriptive statistics of demographic variables and missingness in the main trial outcome at baseline or 3-months. Table D: Sensitivity analysis of primary and secondary outcomes: complete case-predicted means, between-group differences, confidence intervals, and $p$ values ($N=310$). Table E: Sensitivity analysis of primary outcome: complete case-predicted means, between-group differences, confidence intervals, and p values after excluding extreme outliers (based on $1.5 \times$ IQR criterion) ($n=292$). Table F: Sensitivity analysis of primary outcome: multiply-imputed predicted means, between-group differences, confidence intervals, and p values after excluding extreme outliers (based on $1.5 \times$ IQR criterion) ($N=351$). Table G: Observed and predicted median difference in binge drinking days and Wilcoxon rank-sum test results across analytic samples. Table H: Pooled coefficients from the zero-inflated model for the primary binge drinking days outcome, based on 100 multiply imputed datasets ($n=369$). Table I: Pooled coefficients for the drinking days secondary outcome, based on 100 multiply imputed datasets, from models fit using negative binomial, zero-inflated negative binomial, and generalized Poisson distributions ($n=369$). Table J: Bootstrapped within- and between-arm differences in predicted mean change in drinking days, based on 100 multiply imputed datasets, for models fit using negative binomial, zero-inflated negative binomial, and generalized Poisson distributions ($n=369$). Table K: Observed and predicted median difference in drinking days and Wilcoxon rank-sum test results. Table L Pooled coefficients for the drinking amount secondary outcome, based on 100 multiply imputed datasets, from models fit using negative binomial, zero-inflated negative binomial, and generalized Poisson distributions. Table M: Bootstrapped within- and between-arm differences in predicted mean change in drinking amount, based on 100 multiply imputed datasets, for models fit using negative binomial, zero-inflated negative binomial, and generalized Poisson distributions ($n=369$). Table N: Observed and predicted median difference in drinking amount and Wilcoxon rank-sum test results.
(DOCX)

**S2 File. Clinical Trials Gov protocol—PRACT: A pragmatic randomized adaptive clinical trial to investigate controlling alcohol-related harms in a low-income setting; emergency department brief interventions in Tanzania protocol.**
(PDF)

**S3 File. Statistical supplement—Sample size, periodic assessments and citations for R packages used.**
(DOCX)

**S4 File. NIAAA Data Archive (NIAAADA) Data Sharing Plan (DSP).**
(PDF)

**S5 File. Statistical analysis plan.**
(DOCX)

**S6 File. Reporting checklist for randomized trial.** Based on the 2025 CONSORT guidelines.
(DOCX)

**S7 File. Variables used in the imputation process.**
(XLSX)

**S8 File. Metadata document.**
(DOCX)

## Acknowledgments

We are thankful for the staff and patients at the Kilimanjaro Christian Medical Centre in Moshi, Tanzania. The authors were not paid to write this article by a pharmaceutical company or other agency. The content is solely the responsibility of the authors and does not necessarily represent the official views of the National Institutes of Health.

## Author contributions

**Conceptualization:** Catherine A. Staton, Joao Ricardo Nickenig Vissoci, Blandina T. Mmbaga.

**Data curation:** Linda Minja, Mia Buono, Kennedy Ngowi, Joao Ricardo Nickenig Vissoci.

**Formal analysis:** Linda Minja, Joao Vitor Perez de Souza, John A. Gallis, Kennedy Ngowi, Joao Ricardo Nickenig Vissoci.

**Funding acquisition:** Catherine A. Staton, Blandina T. Mmbaga.

**Investigation:** Catherine A. Staton, Kennedy Ngowi, Judith Boshe, Joao Ricardo Nickenig Vissoci, Blandina T. Mmbaga.

**Methodology:** Catherine A. Staton, Joao Vitor Perez de Souza, John A. Gallis, Kennedy Ngowi, Joao Ricardo Nickenig Vissoci.

**Project administration:** Catherine A. Staton, Linda Minja, Joao Vitor Perez de Souza, John A. Gallis, Pollyana Coelho Pessoa Santos, Mia Buono, Francis Sakita, Kennedy Ngowi, Judith Boshe, Ashley J. Phillips, Joao Ricardo Nickenig Vissoci.

**Resources:** Catherine A. Staton.

**Supervision:** Catherine A. Staton, Francis Sakita, Judith Boshe, Joao Ricardo Nickenig Vissoci, Blandina T. Mmbaga.

**Validation:** Linda Minja, Joao Vitor Perez de Souza, John A. Gallis, Kennedy Ngowi, Joao Ricardo Nickenig Vissoci, Blandina T. Mmbaga.

**Visualization:** Linda Minja, Joao Vitor Perez de Souza, Joao Ricardo Nickenig Vissoci.

**Writing – original draft:** Catherine A. Staton, Joao Vitor Perez de Souza, Pollyana Coelho Pessoa Santos, Mia Buono, Ashley J. Phillips, Joao Ricardo Nickenig Vissoci.

**Writing – review & editing:** Catherine A. Staton, Linda Minja, Joao Vitor Perez de Souza, John A. Gallis, Pollyana Coelho Pessoa Santos, Francis Sakita, Kennedy Ngowi, Judith Boshe, Ashley J. Phillips, Joao Ricardo Nickenig Vissoci, Blandina T. Mmbaga.

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
