## [Editor Report · Decision Letter 0]

18 Dec 2024

Dear Dr Staton,

Thank you for submitting your manuscript entitled "Effectiveness of a Brief Intervention and Text-Based Booster to Reduce Harmful and Hazardous Alcohol Use in the Emergency Department of a Low-Resource Setting: A Pragmatic Randomized Adaptive Clinical Trial in Moshi, Tanzania" for consideration by PLOS Medicine.

Your manuscript has now been evaluated by the PLOS Medicine editorial staff and I am writing to let you know that we would like to send your submission out for external peer review.

We also ask that you provide a copy of the original trial protocol and a completed CONSORT checklist as supporting information. By protocol, we mean the complete and detailed plan for the conduct and analysis of the trial that the ethics committee approved before the trial began. Please send this in the original language. If this is in a language other than English, please also provide a translation. Please detail any deviations from this study protocol in the Methods section of your manuscript. The documents will be made available to the editors and reviewers.

Please re-submit your manuscript within two working days, i.e. by Dec 20 2024.

Feel free to email me at atosun@plos.org or us at plosmedicine@plos.org if you have any queries relating to your submission.

Kind regards,

Alexandra Tosun, PhD

Associate Editor

PLOS Medicine

---

## [Decision Letter · Decision Letter 1]

28 Feb 2025

Dear Dr Staton,

Many thanks for submitting your manuscript "Effectiveness of a Brief Intervention and Text-Based Booster to Reduce Harmful and Hazardous Alcohol Use in the Emergency Department of a Low-Resource Setting: A Pragmatic Randomized Adaptive Clinical Trial in Moshi, Tanzania" (PMEDICINE-D-24-04283R1) to PLOS Medicine. The paper has been reviewed by subject experts and a statistician; their comments are included below and can also be accessed here: [LINK]

As you will see, the reviewers raise significant concerns about the reporting, particularly in relation to the trial protocol. We ask that you carefully consider the reviewers' comments. Please note that the trial should be reported according to the original protocol and that there must be sufficient detail throughout the manuscript. After discussing the paper with the editorial team and an academic editor with relevant expertise, I'm pleased to invite you to revise the paper in response to the reviewers' comments. We plan to send the revised paper to some or all of the original reviewers, and we cannot provide any guarantees at this stage regarding publication.

We ask that you submit your revision by Mar 21 2025. However, if this deadline is not feasible, please contact me by email, and we can discuss a suitable alternative.

Don't hesitate to contact me directly with any questions (atosun@plos.org).

Best regards,

Alexandra

Alexandra Tosun, PhD

Associate Editor

PLOS Medicine

atosun@plos.org

Comments from the academic editor:

In addition to the reviewers' comments, I recommend that authors are asked to make the overall design of their adaptive trial clearer to the reader, and to add a few sentences about the value of adaptive RCTs, as they are not familiar to everyone.

Comments from the reviewers:

Reviewer #1: Thank you for the opportunity to review this manuscript. I commend the authors for writing a clear, succinct, and easy to to read manuscript. I have some comments and suggestions.

Major comments

1. Why were the two PPKAY arms (one with standard SMS booster and another with personalized SMS booster) analysed as one group? Of course, I understand that they both received the brief intervention, but why include these two distinct intervention arms if they are not analysed separately? This may be done at a later stage, in another analysis, but its omission here seems ominous.

2. I would expect the baseline binge drinking days, drinking days, number of drinks etc. to be balanced between groups as baseline, and there is no discussion to hypothesise why this is not the case? Is it a failure of randomization, too small sample size?

3. Given the open label nature of the intervention and self reporting of the outcome, there is a big risk of detection bias; those in the intervention arm are more likely to self-report lower alcohol use. The authors state in the limitations that they do not believe the self-reporting practices should differ by intervention arm, but there is no empirical evidence to prove this. One way to address this is to have a negative control outcome, that shares similar determinants of detection as the outcome of interest, but is not expected to be affected by the intervention. For example, this could be something like number of days eating junk food in the last 3 months? Did the authors collect any outcomes they could use as a negative control outcome?

4. Looking at the results in Table 2, the median binge drinking days is 0 for all groups both at baseline and at 3 months. Similarly, for the other outcomes the median values are very low. Across all outcomes, the mean is considerably larger than the median, which suggests the data are considerably skewed. A way to address this problem would be to define the outcomes as the proportion of people above a certain clinically relevant threshold. For example, the proportion of people with greater than 3 binge drinking days in the last 3 months. The addition of analyses with outcome definitions like above, perhaps as secondary analyses, would greatly improve the paper.

Minor comments

1. There are too many acronyms used throughout the manuscript, which makes it hard to read in places

2. Were all the nurses trained on how to deliver the PPKAY? And was that training standardized?

3. How many replications were used in the bootstrapping process?

4. Why are the limitations positioned after the conclusions? Surely the conclusions should be the last part of the manuscript?

5. P-values do not need to be reported to 4 decimal places. 2 is sufficient.

Reviewer #2: This manuscript describes the results of an adaptive randomized trial of a harmful alcohol reduction intervention conducted in Tanzania. The trial has an interesting design, and some of the findings support the intervention. I do have some major concerns and clarifying questions about how the analysis was conducted and how results are currently reported.

Major concerns:

1. The primary outcome, as reported in the manuscript, differs from the pre-specified primary outcome in the protocol. The protocol specifies that DrInC (here presented as a secondary outcome that was not statistically significant) is the primary outcome, and specifies binge drinking as one of a few secondary outcomes. Was a protocol revision made prior to unblinding that changed the primary outcome? If not, this manuscript needs to transparently report the original primary outcome and secondary outcomes and not switch them.

2. Outcomes are self-reported. Authors indicate in the discussion that this could be subject to social desirability bias, but that these biases would balance across arms. I don't necessarily agree. Participants were not blinded to the intervention they received, correct? So those who received interventions related to alcohol harm reduction might be more likely to report less harmful alcohol behavior, particularly in a personal interview setting. If this type of differential social desirability in reporting occurred, the intervention effect would be arbitrarily inflated.

3. I have several concerns about the analysis conducted and how results are reported.

a. The authors need to clearly state which analyses were prespecified and which were not. For example, the protocol does not specify the use of zero inflated modeling. Was this prespecified in the SAP (which wasn't included) or changed after analysis began? Were the covariates included in models pre-specified somewhere?

b. The authors pre-specified in the protocol that they would conduct multiple imputation for missing data, and they did not. There is a substantial amount of missing data in both arms (29% usual care, 32% intervention arms), so complete case analysis alone is not sufficient. For all complete case analyses, results need to be caveated by the n, % of missing data, which they currently don't report clearly (it can be gleaned from Figure 1). But consistent with the protocol, the primary analysis should include MI.

c. The analysis methods seem unnecessarily complicated (and introduce unnecessary parametric assumptions). For each outcome, the authors could just compute the change in scores or counts for participants from baseline to the end of study and conduct a nonparametric test for these change scores (e.g., Wilcoxon rank sum test). Corresponding estimates of median difference between arms with 95% CIs can be reported. I recommend this be conducted as a sensitivity analysis, as the inclusion of so many baseline covariates and parametric assumptions make the current results difficult to interpret.

d. For each model, authors need to say what covariates were included and how the "predicted means" were computed. For zero-inflated models, there is a model for both the probability of being an excess zero and a model for the mean count given that one is not an excess zero. Covariates in each need to be described. How was time incorporated in each longitudinal model (e.g., as a linear term)? In general, authors should say what variables were included in each model and how overall estimates were obtained from the models. A detailed statistical appendix would improve clarity/interpretability of all analyses.

e. It is not appropriate to conduct statistical testing for baseline/Table 1 variables.

4. The sample size calculations differ from the protocol, as those were based on a different primary outcome. Which was used to power the study? Authors also need to say what statistical test was used to determine power and exactly what sample size was needed according to the calculations.

Other comments:

1. This manuscript pools together two intervention arms. This is prespecified in the protocol and part of the adaptive design, but this pooling will likely confuse readers. Authors should explain why this was done early on to avoid confusion.

2. The incidence rate reported in the second sentence of the introduction needs to be in terms of some unit of time, for it to be a rate.

3. Authors talk about using the bootstrap to "estimate the confidence intervals". We only estimate parameters, and confidence intervals are not parameters. Instead, they could say "compute" or "construct".

4. Table 2 would be more interpretable if mean/median differences in change scores between arms were included, with 95% CIs. This would parallel nicely with the sensitivity analysis I recommended above.

5. Figure 2 is overly complicated. The bars just indicate where the means are and could be removed. A more meaningful figure might depict individual level data, like via a scatterplot in the background of the bars.

6. In the limitations, authors refer to "a regression to the mean" but don't explain what they mean by this or cite other literature for the "As in many alcohol trials" claim.

Reviewer #3: This article reports the results of a superiority clinical trial comparing the effectiveness of a culturally adapted brief intervention with mobile health-based boosters to usual care. Participants were adults presenting to an Emergency Department in Tanzania for an acute injury and either reporting alcohol use prior to the injury, Alcohol Use Disorder Identification Test ≥8, and/or having positive BAC according to breathalyzer measures. Participants were randomly assigned to 2 intervention conditions (ED brief intervention + personalized mobile booster, and ED brief intervention + standard mobile booster) or usual care, with a 1:1:1 allocation ratio. The authors pooled the 2 intervention arms and compared them to usual care to determine the effectiveness of the intervention in reducing alcohol use and consequences at 3-month follow-up. Analyses showed statistically significant differences for 3 alcohol use measures, but no significant differences were found between the two groups in terms of drinking-related consequences or depression.

The topic is of interest and testing the effects of a culturally adapted intervention approach is both timely and well-founded. Nevertheless, there are several methodological concerns and lack of design description resulting in skepticism about finding reliability and relevance. These major concerns are described below, followed by minor points listed in order of appearance in the text.

First of all, it is not clear why the design have 3 arms, 2 of which are pooled in the analysis. I think I understand after having read the study protocol, but neither the introduction nor the methods section clearly explains this complicated design. As such, the study design looks wrong (one should not combine 2 groups when randomization is conducted 1:1:1, at least not as the primary analysis).

According to the study protocol, other endpoints were collected (6, 9, 12 and 24 months), but this paper only report 3-month outcomes. The publication strategy is not clear and should be described in the introduction. Without a more detailed explanation, it might look like a "salami slicing" strategy.

The choice of the primary outcome is questionable. The median is 0 at baseline and the 25th and 75th percentile are 0 at follow-up. The standard deviations also indicate a strongly skewed distribution. Why using such a rare behavior as the main outcome? It raises also the question of participants selection. Were the inclusion criteria too wide?

Related to this, the main results are complicated to understand. Looking at medians, there were no changes across time among both groups (0 to 0 in both groups). Looking at means, there were important decreases in both groups (2.2 to 0.8 in the control group and 3.0 to 0.4 in the intervention group). Looking at regression models, there is a decrease of 3.4 (95% CI -4.8; -2.3). How is this possible? How is such a 95% CI possible with both medians at 0? An explanation is that most participants did not change (since they could not decrease as they did not binge…), but a small group did decrease and draws all the effect. This should be better described (and maybe tested in secondary analyses). The findings in the abstract should also be revised since currently only the important effect of the regression model is reported.

MINOR POINTS

Abstract

The three conditions should be better described in the abstract.

Authors should indicate participation rate (448 over how many eligible?) in the Findings section.

The sentence reporting the main effect is not correct. The difference reported is the difference in difference and not the reduction among the intervention group per se. See also, my comment above about the findings being reported.

Introduction

"Alcohol use is a significant global health crisis". This sentence should be rephrased.

Authors should add a reference when describing BI as incorporating MI.

The literature on BI in the ED is restricted to a few studies. There are several other studies, as well as literature reviews on this topic. There are also other studies testing the effect of models conducted by addiction liaison staff. And other studies testing the effects of booster sessions.

Methods

How was the binge drinking days outcome measured? Timeline follow-back procedure? Or a single self-report question? Same question for number of standard drinks. It would be difficult to remember number of drinks over the last 30 days without a TLFB procedure.

Consider renaming "alcohol use behavior" as measured by the AUDIT. Either keep "AUDIT score" or state something like "AUD severity".

The sample size estimation is poorly presented in the methods. Additional information should be provided to justify the expected size of effect and the corresponding numbers.

Why did the authors use different approaches for drinking and non-drinking outcomes. A baseline adjustment approach would have been possible for AUDIT, DRINC, and PHQ outcomes. Why using mixed models here and not for drinking outcomes? In the protocol, baseline adjusted approach was proposed for all analyses.

Results

The first sentence of the Results is unclear. Authors did conduct a second interim look for 2 years and a half? This sentence should be rephrased and the use of interim analyses and stopping rules should be presented in the methods (and maybe introduced in the introduction).

The patient flow should be more detailed. What were the causes of exclusion of the 1032 not included? What are the "Other" reason of not completing follow-up? How come that staff did not attempt to contact 4 participants in the intervention group?

Discussion was not reviewed since there were too many issues in the introduction and methods, preventing reliable and interpretable findings.

Reviewer #4: C1. Thank you for the opportunity to review this manuscript evaluating the effectiveness of brief intervention plus text-based messages to reduce harmful and hazardous alcohol use in the emergency department. Many aspects of this work need to be clarified to enhance the reporting of both methods and results and to enhance the overall relevance of this work to the journal's readers.

For example, it is unclear if recruited participants in the emergency department worked with one nurse or multiple nurses for the entirety of the brief intervention and SMS Booster messages. If multiple, how can authors ensure the fidelity of the intervention?

C2. The lack of details on training the nurses and research assistants to ensure the delivery of a standardized intervention is unclear.

C3. The authors need to discuss in detail the possible response bias introduced by having unblinded research nurses open the SNOSE to determine group assignment, conduct the recruitment, and implement the intervention.

C4. There are considerable concerns given the reliance on self-reporting for primary and secondary outcomes. For example, the standardized AUDIT questions assess alcohol use within one year, and it is not appropriate to change the questions' timeframe to a three-month time window. The authors also need to explain why they used the number of binge drinking days as the primary outcome instead of alcohol consumption in grams per week. The authors need to check the core outcome set (COS) recommended by the International Network on Brief Interventions for Alcohol and Other Drugs (INEBRIA) organization. The COS covers domains of average alcohol consumption (e.g., alcohol consumption in grams per week), frequency of alcohol consumption (e.g., frequency of drinking in past 30-day), and episodes of hazardous or harmful drinking (e.g., episodes of binge drinking and heavy drinking in past 30-day). The COS followed the Core Outcome Measures in Effectiveness Trials (COMET) guidelines and recommended the evaluation of the effectiveness of alcohol trials in all settings.

C5. The sample size estimation needed to be written in detail and the calculation did not account for the possible retention rate at the follow-up and the allocation ratio.

C6. The authors stated that the primary outcome analysis was based on intention-to-treat. However, the number of participants included for primary outcome analysis in the Figure 1 flowchart did not say so.

C7. The authors did not perform sensitivity analyses (a requirement of the CONSORT guideline for RCT study) to handle the missing data.

C8. It is unclear from the results if there was a dose-response between the number of SMS messages a participant read and reductions in the number of binge drinking days, standard drinks, and drinking days.

C9. The authors need to state the theoretical framework that guided the design of the SMS booster messages to increase the study's transferability.

C10. The authors did not mention any details on the usual care practice.

C11. The authors stated that this is a pragmatic RCT. It is unclear which components of this study's design were pragmatic in nature. Clarification is needed, including using the CONSORT extension to report on pragmatic trials and enhance transparency.

C12. The authors did not exclude the potential participants with psychiatric disorders or participating in other alcohol prevention and treatment programme. For example, there is a bi-directional association between alcohol consumption and mental illness that might considerably affect the study outcomes.

C13. The authors did not mention how to handle the potential confounders when performing data analyses in the methodology nor include them in the footnote of the tables. Although it is an RCT study, the confounder still need to be adjusted. Please check the reference below.

Holmberg MJ, Andersen LW. Adjustment for Baseline Characteristics in Randomized Clinical Trials. JAMA. 2022 Dec 6;328(21):2155-2156.

C14. The author did not mention that any incentive would be given to the participants who completed the follow-up. Details on compensation or what was driving intervention engagement need to be thoroughly discussed and assessed.

---

* Please upload any figures associated with your paper as individual TIF or EPS files with 300dpi resolution at resubmission; please read our figure guidelines for more information on our requirements: http://journals.plos.org/plosmedicine/s/figures. While revising your submission, please upload your figure files to the PACE digital diagnostic tool, https://pacev2.apexcovantage.com/. PACE helps ensure that figures meet PLOS requirements. To use PACE, you must first register as a user. Then, login and navigate to the UPLOAD tab, where you will find detailed instructions on how to use the tool. If you encounter any issues or have any questions when using PACE, please email us at PLOSMedicine@plos.org.

* FINANCIAL DISCLOSURES: The funding statement should include: specific grant numbers, initials of authors who received each award, URLs to sponsors’ websites. Also, please state whether any sponsors or funders (other than the named authors) played any role in study design, data collection and analysis, the decision to publish, or preparation of the manuscript. If they had no role in the research, include this sentence: “The funders had no role in study design, data collection and analysis, decision to publish, or preparation of the manuscript.”

* COMPETING INTERESTS: All authors must declare their relevant competing interests per the PLOS policy, which can be seen here: https://journals.plos.org/plosmedicine/s/competing-interests

* DATA AVAILABILITY: The Data Availability Statement (DAS) requires revision. For each data source used in your study:

FIGURES AND TABLES

SUPPLEMENTARY MATERIAL

REFERENCES

STUDY TYPE-SPECIFIC REQUESTS

* PLOS Medicine requires that all trials be prospectively registered in one of registries recognized by WHO. Please ensure that study registration details are included in the Methods section.

* Please structure the Methods section using the following sub-headings: Study design and participants, Randomization and masking, Procedures, Outcomes, Statistical analysis.

* Please ensure that all outcomes measures are reported according to the trial protocol [and/or trial registry]. Please clarify and explain all discrepancies between the paper and protocol. If the outcomes were not prespecified in the protocol, please define them in the Methods (Outcomes section) as post hoc and explain why they were added. Post-hoc comparisons should be presented as hypothesis generating rather than conclusive.

* Please ensure that all prespecified outcomes (primary, secondary, and exploratory) are listed in the Methods/Outcomes section and indicate whether there are outcomes that are not presented in the current report.

* Please specify the dates (Month Day, Year) during which study enrollment and follow up occurred.

* Please include absolute numbers wherever you report percentages; eg, n/N (%)

* Please present the safety data for the study including numbers of specific events and whether or not adverse events are thought to be related to treatment. AEs should be reported in the abstract, per CONSORT and CONSORT-Harms.

* Please complete the CONSORT checklist (https://www.equator-network.org/reporting-guidelines/consort/) and ensure that all components of CONSORT are present in the manuscript, including how randomization was performed, allocation concealment, blinding of intervention, definition of lost to follow-up, power statement. When completing the checklist, please use section and paragraph numbers, rather than page numbers.

* Please report your abstract according to CONSORT for abstracts, following the PLOS Medicine abstract structure (Background, Methods and Findings, Conclusions) https://www.equator-network.org/reporting-guidelines/consort-abstracts/

* If your trial had to undergo important modifications in response to extenuating circumstances, please complete the CONSERVE-CONSORT checklist and provide in your Supporting Information; (https://www.equator-network.org/reporting-guidelines/guidelines-for-reporting-trial-protocols-and-completed-trials-modified-due-to-the-covid-19-pandemic-and-other-extenuating-circumstances-the-conserve-2021-statement/). When completing the checklist, please use section and paragraph numbers, rather than page numbers.

* In keeping with our commitment to Open Science, please include the study protocol document and analysis plan (including any amendments) as Supporting Information to be published with the manuscript if accepted.

* Please note that PLOS Medicine requires prospective, public registration of a data sharing plan (as part of mandatory clinical trials registration) for all clinical trials that began enrollment on or after January 1, 2019, in accordance with ICMJE requirements.

---

## [Decision Letter · Decision Letter 2]

8 Jul 2025

Dear Dr Staton,

Many thanks for re-submitting your manuscript "Effectiveness of a Brief Intervention and Text-Based Booster to Reduce Harmful and Hazardous Alcohol Use in the Emergency Department of a Low-Resource Setting: A Pragmatic Randomized Adaptive Clinical Trial in Moshi, Tanzania" (PMEDICINE-D-24-04283R2) to PLOS Medicine. The paper has been seen again by a subject expert and a statistician; their comments are included below and can also be accessed here: [LINK]

As you will see, the reviewers still have concerns about the manuscript. After discussing the paper with the editorial team, we invite you to thoroughly address the comments in a further revision. We plan to send the revised paper to some or all of the original reviewers, and we cannot provide any guarantees at this stage regarding publication.

We ask that you submit your revision by Jul 29 2025. However, if this deadline is not feasible, please contact me by email, and we can discuss a suitable alternative.

Don't hesitate to contact me directly with any questions (atosun@plos.org).

Best regards,

Alexandra

Alexandra Tosun, PhD

Senior Editor

PLOS Medicine

atosun@plos.org

Comments from the editorial team:

Please note that, for further consideration, it is essential to address the reviewers' concerns to their satisfaction.

The editorial team agrees with Reviewer #2 that the self-reported outcome should be presented as a major limitation in both the Discussion and the Abstract. In addition, when revising the manuscript, please thoroughly document the statistical methods in a way that would allow others to replicate your findings.

Comments from the reviewers:

Reviewer #2: This manuscript has been improved from the prior version. However, many of my original concerns remain, and there are a few additional that need clarification.

Major concerns:

1. Self-reported outcomes: As reiterated by multiple reviewers, there is the potential for social desirability bias, and it could be differential due to non-blinding of participants. The manuscript still says in the limitations, "Social desirability may have influenced overall reporting levels, but such bias is likely to have been similar across both arms." There is no basis for this. The fact that secondary endpoints were not statistically significant doesn't mean there was no social desirability bias in the primary endpoint. Due to its potential for impacting the primary findings, this limitation needs to be clearly acknowledged in the limitations section and, I would argue, the abstract.

2. Missing Data/Multiple imputation: MI was pre-specified in the protocol, and thus should be the primary analysis presented in the main text, not as a sensitivity analysis. 10 imputations seems low to me - can the authors explain how this was chosen? Furthermore, the MI approach should result in an estimate of the same endpoint as is currently presented for the primary outcome, not just a table of model coefficients as in Table S4. For the description of missing data, the authors conclude that missingness is consistent with a MAR mechanism. MNAR cannot be distinguished from MAR in tables of observed covariates, so this statement is much too strong. They can say instead that MCAR was ruled out and that they will assume MAR based on a given set of covariates. Covariates used in the MI model need to be listed.

3. Sensitivity Analysis: I maintain that the analytic methods are unnecessarily complicated and introduce unnecessary parametric assumptions. They are also susceptible to overinterpreting the intervention due to outliers. I recommend that the authors do include the more straightforward, less assumption dependent, and less prone to the influence of outliers Wilcoxon results (median, 95% CI) as an added sensitivity analysis.

4. Detailed statistical methods need to be described. For each model, authors need to say what covariates were included. For zero-inflated models, there is a model for both the probability of being an excess zero and a model for the mean count given that one is not an excess zero. Covariates in each need to be described. How was time incorporated in each longitudinal model (e.g., as a linear term)? In general, authors should say what variables were included in each model and how overall estimates were obtained from the models. As previously suggested, a detailed statistical appendix would improve clarity/interpretability of all analyses. The current appendix is just a list of R packages, and the SAP itself does not provide these details. The SAP even includes a different endpoint than is reported in the paper (odds ratios).

5. Influence of outliers: I share a concern voiced by other reviewers. The difference could potentially be driven by a few outliers. To see if this is the case, I recommend four things. (1) add the proportion of participants with 1+ binge drinking days to table 2, (2) add the max for each outcome in table 2 (in addition to Q1/Q3/mean/median), (3) in Figure 2, as previously recommended, add individual level data, like via a scatterplot in the background of the bars, and (4) include the sensitivity analysis recommended above.

6. Use of p-values: The "statistical significance" language needs to be toned down. Per guidance from the American Statistical Association's statement on the recommended use of p-values (https://journals.lww.com/joacp/fulltext/2016/32040/the_american_statistical_association_statement_on.1.aspx), I recommend only discussing statistical significance for outcomes that were powered. For secondary outcomes, the "no significant differences" language should be removed, and instead a meaningful comparison of differences with CIs should be discussed. Furthermore, statistical testing should not be conducted for missing data analysis. The study was not powered for these comparisons, so p-values are not meaningful.

7. Participant Drop out. Participants should state in the text that data for the primary outcome was only available for 68% of intervention participants and 71% in usual care so that readers don't have to calculate this from Table 1.

Other comments:

1. In the response to reviewer 2, comment 3c, the authors say that "However, we had defined to conduct a more robust model that would allow us to estimate the more accurate estimate adjusting for baseline confounders." I would emphasize that there is no confounding in a truly randomized clinical trial (unless you mean informative missing data). Adjustments for baseline covariates are for the purposes of efficiency gain, not bias reduction.

2. In author summary, authors should summarize both significant findings as well as secondary outcomes that were similar between arms.

Reviewer #3: The authors made a great work to answer reviewers' points. Most points were adequately treated (in particular presentation of the design) and the resulting manuscript is highly enhanced.

There are however a few important points that remain unclear.

1. It is not clear how the authors handled time in their analyses. It looks like they used both baseline adjustment and change scores (difference between T1 and T2). If authors used a change score, they should not adjust this for baseline value, since this information would be used twice.

2. The "difference in differences" are sometimes highy significant while the confidence intervals include 0. How is this possible? Either results are incorrect or it raises high doubts the quality of the data.

3. The flow chart should include detials on exclusion criteria.

---

## [Decision Letter · Decision Letter 3]

12 Sep 2025

Dear Dr. Staton,

Thank you very much for re-submitting your manuscript "Effectiveness of a Brief Intervention and Text-Based Booster to Reduce Harmful and Hazardous Alcohol Use in the Emergency Department of a Low-Resource Setting: A Pragmatic Randomized Adaptive Clinical Trial in Moshi, Tanzania" (PMEDICINE-D-24-04283R3) for review by PLOS Medicine.

Thank you for your detailed response to the reviewers' and editors’ comments. I have discussed the paper with my colleagues and an academic editor with relevant expertise, and it has also been seen again by two of the original reviewers. The changes made to the paper were mostly satisfactory to the reviewers. As such, we intend to accept the paper for publication, pending your attention to the reviewers' and editors' comments below in a further revision. When submitting your revised paper, please once again include a detailed point-by-point response to the reviewers' and editorial comments.

The remaining issues that need to be addressed are listed at the end of this email. Any accompanying reviewer attachments can be seen via the link below. Please take these into account before resubmitting your manuscript: [LINK]

In revising the manuscript for further consideration here, please ensure you address the specific points made by each reviewer and the editors. In your rebuttal letter you should indicate your response to the reviewers' and editors' comments and the changes you have made in the manuscript. Please submit a clean version of the paper as the main article file. A version with changes marked must also be uploaded as a marked up manuscript file. Please also check the guidelines for revised papers at http://journals.plos.org/plosmedicine/s/revising-your-manuscript for any that apply to your paper.

We ask that you submit your revision within 1 week (Sep 19 2025). However, if this deadline is not feasible, please contact me by email, and we can discuss a suitable alternative.

Please do not hesitate to contact me directly with any questions (atosun@plos.org).

We look forward to receiving the revised manuscript.

Sincerely,

Alexandra Tosun, PhD

Senior Editor 

PLOS Medicine

plosmedicine.org

Comments from Reviewers:

Reviewer #2: The authors have done an excellent job responding to the previous reviews. My only comment is that it would be good to provide a high-level summary of the sensitivity analyses for the primary outcome in the main text rather than just pointing to them in the supplement.

Reviewer #3: The revisions carried out by the authors have satisfactorily addressed reviewers' comments.

Here are a few last minor points.

In the discussion, the sentence « Our study demonstrates the effectiveness of a culturally adapted and pragmatic implementation of a brief intervention with specific intervention fidelity focus for injury patients" should be toned down, taking into account study limitations and design.

Same comment for the first sentence of the Conclusions.

The number of decimals in some reported numbers is unusually long. Refer to journal guidelines, but numbers (e.g. p values) are usually rounded to two decimals.

As the adaptative design may be less familiar to many readers, the conclusions of the abstract and the conclusions of the article should mention further stages of the adaptative design.

[LINK]

Requests from Editors:

GENERAL

* Please note that we require authors to include page and line numbers in their manuscript for review purposes.

* Please confirm that your title complies with to PLOS Medicine's style. Your title must be nondeclarative and not a question. It should begin with main concept if possible. "Effect of" should be used only if causality can be inferred, i.e., for an RCT. Please place the study design ("A randomized controlled trial," "A retrospective study," "A modelling study," etc.) in the subtitle (i.e., after a colon).

Suggestion: Effectiveness of a Brief Intervention and Text-Based Booster in the Emergency Department to Reduce Harmful and Hazardous Alcohol Use: A Pragmatic Randomized Adaptive Clinical Trial in Moshi, Tanzania

* Statistical reporting: Please revise throughout the manuscript, including tables and figures.

- Please report statistical information as follows to improve clarity for the reader ""XX (95% CI [XX,XX]; p</=)"".

- Please separate upper and lower bounds with commas instead of hyphens as the latter can be confused with reporting of negative values.

- Please repeat statistical definitions (HR, CI etc.) for each set of parentheses.

* Please ensure that all abbreviations are defined at first use throughout the text (including statistical abbreviations). Please also check figures and tables.

* Please ensure that tables and figures, including those in supplementary files, are appropriately referenced in the main text.

* Please check that any use of statistical terms (such as trend or significant) are supported by the data, and if not please remove them.

* The Data Availability Statement (DAS) following the main text (“Data Sharing”) and the DAS in the online submission form differ. Please check and ensure to provide the correct DAS in the online submission form. Please note that PLOS Medicine requires that the de-identified data underlying the specific results in a published article be made available, without restrictions on access, in a public repository or as Supporting Information at the time of article publication, provided it is legal and ethical to do so.

* Please revise for use of patient-centered language. Please note that patient-centered language is constructed with the use of post-modified nouns (e.g. 'patients with injury’ (or similar) instead of ‘injury patients’) putting the person first in the sentence structure.

ABSTRACT

* Please confirm that your abstract complies with our requirements, including providing all the information relevant to this study type https://journals.plos.org/plosmedicine/s/submission-guidelines#loc-abstract

* Per CONSORT, please note that only the primary outcome of the trial should be reported in your Abstract. Secondary outcomes should only be included in the Abstract if all secondary outcomes are fully reported. For trials that have many secondary outcomes, the Abstract should be limited to reporting the primary outcome.

* “We report the results of a superiority clinical trial comparing the effectiveness of a culturally adapted brief intervention, “Punguza Pombe Kwa Afya Yako” (PPKAY, Reduce Alcohol for your Health), with text-based boosters to usual care.” – Please note that the final sentence should clearly state the study question.

* “This manuscript will report…” – please change tense.

* Please clearly state the primary outcome of the trial.

* Please include years during which the study took place.

* Please state that analysis was intention to treat.

* Please provide the number of participants lost to follow up in each group.

* Please include baseline characteristics of the study population (e.g. mean age, sex).

* Please ensure that all numbers presented in the abstract are present and identical to numbers presented in the main manuscript text.

* Please change ‘Interpretation’ to ‘Conclusions’.

AUTHOR SUMMARY

* In the author summary, in the final bullet point of 'What Do These Findings Mean?', please include the main limitations of the study in non-technical language (as a separate bullet point).

* We suggest removing the information on secondary outcomes and focusing only on the primary outcome. This would streamline the author summary.

METHODS AND RESULTS

* Thank you for providing your CONSORT checklist. Please replace the page numbers with paragraph numbers per section (e.g. "Methods, paragraph 1"), since the page numbers of the final published paper may be different from the page numbers in the current manuscript. Also, please ensure to use the CONSORT 2025 checklist.

* “…but will be presented in brief below and in the Supplementary Information (S2 File).” – please ensure to add clear labels to the files. Currently, there’s no file labelled as S2 file.

* Since the trial focuses on the PPKAY intervention, we suggest providing more information about the intervention (e.g., under "Brief Intervention") rather than relying on references 19 and 26. We want readers to find information about the cultural adaptations in the main text.

* Please confirm that the primary and secondary outcomes are reported in accordance with the trial protocol, and indicate any changes in the main text, if applicable.

* Please include the statement on code in the data availability statement.

* Please confirm that changes in the analysis-- including those made in response to peer review comments—have been identified as such in the Methods section of the paper, with rationale. If not, please revise accordingly.

* You state that the current manuscript presents the second interim analysis with 65% enrollment. However, according to your sample size calculation, the second interim analysis should include approximately 320 participants. However, it seems that you had already included 448 participants by that time. Please comment and clarify.

* “The primary and secondary outcome results presented in this study represent data imputed using mice.” – please capitalize ‘mice’. If used for the first time, please define the abbreviation.

* Table 1/2: Please clarify that the intervention group is a combination of personalized and standard booster messages.

* “Across all evaluations, results were directionally consistent, with greater reductions in binge drinking days observed in the intervention group.” – based on Figure S1, we recommend toning down this statement and providing a more accurate description of the results.

* Figure 2: Why aren't you using a box plot with single data points, as shown in Figure S1? We believe that box plots are more informative and transparent. Also, when a p-value is provided, please include the statistical test used to determine it in the legend.

* Table 3: When a p value is given, please specify the statistical test used to determine it in the legend.

* Due to the large standard deviations for the primary and secondary outcomes and the rarity of the outcome(s), clearly state that there is a high degree of uncertainty.

* Tables: Please specify the variables controlled for in all relevant Tables.

DISCUSSION

* Pleas remove all subheadings, including the ‘Conclusion’ subheading.

* When revising the discussion, please consider toning down any conclusions given the rarity of your primary outcome and the lack of a comparator group.

* ”Our pragmatic randomized adaptive clinical trial has demonstrated effectiveness of our culturally- adapted brief intervention, Punguza Pombe Kwa Afya Yako (PPKAY) with mobile health-based boosters reduces, on average, 1.2 binge drinking days per 4 weeks compared to usual care.” – We believe this should say: “Our pragmatic, randomized, adaptive clinical trial demonstrated the effectiveness of our culturally adapted brief intervention, Punguza Pombe Kwa Afya Yako (PPKAY). Compared to usual care, PPKAY with mobile health-based boosters reduced binge drinking by an average of 1.2 days more per four weeks.”

General Editorial Requests

---

## [Editor Report · Decision Letter 4]

29 Sep 2025

Dear Dr Staton, 

On behalf of my colleagues and the Academic Editor, Charlotte Hanlon, I am pleased to inform you that we have agreed to publish your manuscript "Effectiveness of a Brief Intervention and Text-Based Booster in the Emergency Department to Reduce Harmful and Hazardous Alcohol Use: A Pragmatic Randomized Adaptive Clinical Trial in Moshi, Tanzania" (PMEDICINE-D-24-04283R4) in PLOS Medicine.

I appreciate your thorough responses to the reviewers' and editors' comments throughout the editorial process. We look forward to publishing your manuscript, and editorially there are only a few remaining points that should be addressed prior to publication. We will carefully check whether the changes have been made. If you have any questions or concerns regarding these final requests, please feel free to contact me at atosun@plos.org.

Please see below the minor points that we request you respond to:

* Figure 2: In the figure description, please define all elements of box plots in the figure caption - center line, box limits and whiskers.

* Please note that we updated the Data Availability Statement in the metadata to include the statement on code.

* “The confidence intervals for both secondary outcomes suggest that the data supports a wider range of intervention effects, from a stronger to a modest effect on drinking frequency and quantity, though the range of plausible effects remains wide.” - In trials, there is usually a distinction in the language in terms of causal vs associational for primary and secondary trial outcomes. It would be beneficial to use associational language in the discussion and other sections for secondary outcomes. Please revise throughout.

Before your manuscript can be formally accepted you will need to complete some formatting changes, which you will receive in a follow up email (including the editorial requests above). Please be aware that it may take several days for you to receive this email; during this time no action is required by you. Once you have received these formatting requests, please note that your manuscript will not be scheduled for publication until you have made the required changes.

PRESS

Sincerely, 

Alexandra Tosun, PhD 

Senior Editor 

PLOS Medicine